# The Connection between MicroRNAs and Oral Cancer Pathogenesis: Emerging Biomarkers in Oral Cancer Management

**DOI:** 10.3390/genes12121989

**Published:** 2021-12-15

**Authors:** Ciprian Osan, Sergiu Chira, Andreea Mihaela Nutu, Cornelia Braicu, Mihaela Baciut, Schuyler S. Korban, Ioana Berindan-Neagoe

**Affiliations:** 1Research Center for Functional Genomics, Biomedicine and Translational Medicine, “Iuliu Hatieganu” University of Medicine and Pharmacy, 400337 Cluj-Napoca, Romania; cipri.osan@yahoo.com (C.O.); sergiu.chira@umfcluj.ro (S.C.); nutu.andreea@umfcluj.ro (A.M.N.); cornelia.braicu@umfcluj.ro (C.B.); 2Department of Maxillofacial Surgery and Implantology, “Iuliu Hatieganu” University of Medicine and Pharmacy, 400033 Cluj-Napoca, Romania; mbaciut@umfcluj.ro; 3Department of Natural Resources & Environmental Sciences, University of Illinois at Urbana-Champaign, Urbana, IL 61801, USA; korban@illinois.edu

**Keywords:** oral cancer, oncomiRs, tumor suppressor miRs, tumor microenvironment, biomarkers

## Abstract

Oral cancer is a common human malignancy that still maintains an elevated mortality rate despite scientific progress. Tumorigenesis is driven by altered gene expression patterns of proto-oncogenes and tumor-suppressor genes. MicroRNAs, a class of short non-coding RNAs involved in gene regulation, seem to play important roles in oral cancer development, progression, and tumor microenvironment modulation. As properties of microRNAs render them stable in diverse liquid biopsies, together with their differential expression signature in cancer cells, these features place microRNAs at the top of promising biomarkers for diagnostic and prognostic values. In this review, we highlight eight expression levels and functions of the most relevant microRNAs involved in oral cancer development, progression, and microenvironment sustainability. Furthermore, we emphasize the potential of using these small RNA species as non-invasive biomarkers for the early detection of oral cancerous lesions. Conclusively, we highlight the perspectives and limitations of microRNAs as novel diagnostic tools, as well as therapeutic models.

## 1. Introduction

Malignant diseases continue to present intriguing challenges for the scientific community as they persistently incur elevated mortality, despite all the tremendous efforts undertaken for pursuing novel and more efficient treatments. Approximately 10 million cancer deaths have been reported in 2020, and this bleak outcome can be associated with cancer cells’ complexity, genetic heterogeneity, and plasticity [1]. Even though surgery and chemo/radiotherapy remain the main approaches for treatment and management of malignant diseases, tumor relapse still poses a challenge, in addition to adverse effects of therapy [2,3].

Oral cancer is a malignancy that affects both the lip and the oral cavity. Its localization sites vary from salivary glands to lymphoid tissues, but the most frequent subtype originates in squamous cells of the oral mucosa [4]. Epidemiologically, there are more than 377,713 new cases diagnosed annually accounting for almost 2% of all sites. Moreover, it has been observed that no less than 177,757 individuals are victims of this malignant disease in 2020; thus, oral cancer is ranked in the top 20 most pernicious types of cancer [1]. This incidence is persistent not only because there is a considerable probability of emerging multiple tumors within 5 years after the initial treatment, but the onset of oral cancer is often insidious as it is masked by anatomical elements of the oral cavity [5]. Hence, there is a high interest in discovering new reliable biomarkers capable of early detection of neoplastic events that will offer advantages in fighting against this life-threatening malignancy [6].

Cancer, as a disease, is associated with altered gene expression levels and function, due to genetic and epigenetic modifications, and it is associated with high genomic instability. Accumulation of such alterations can eventually lead to the development of a malignant phenotype, anarchic proliferation of cells, and metastasis. More specifically, tumor-suppressors genes are inactivated, while proto-oncogenes are activated [7].

MicroRNAs (miRNAs) are short (22–23 nucleotides), non-coding RNAs, first described in *Caenorhabditis elegans* [8]. These unique RNA molecules play critical roles in regulating gene expression, and, consequently, are involved in large numbers of processes in mammals [9,10]. This particularity is attributed to the ability of miRNAs to pair with specific messenger RNAs (mRNAs) and to silence them through an RNA-Induced Silencing Complex (RISC) [11]. According to miRGate (https://tools4mirs.org/software/mirna_databases/mirgate/, accessed on 6 November 2021), 2680 mature miRNAs target more than 60% of protein-coding genes [12,13]. Taking that into consideration, it can be strongly affirmed that dysregulation of miRNAs may contribute to the development of diverse diseases such as cancer [14], liver cirrhosis [15], Parkinson’s disease [16], epilepsy, Alzheimer [17], and many others.

Cancer cells are characterized by loss of homeostasis, and inherently altered regulatory pathways. Therefore, it is not surprising why miRNAs are included, among other factors, in the regulatory mechanism influencing the onset and development of cancer. Altered levels of miRNA expression are highlighted in several neoplastic diseases including cervical [18], colon [19], lung [20], ovarian [21], and breast cancers [22]. Recent studies [23,24,25] have reported that oral cancers are not an exception to this rule but altered levels of miRNAs can lead to carcinogenesis (Figure 1).

The purpose of this review is to highlight the essential roles of miRNA dysregulation in oral cancer development and progression, presenting the most relevant transcripts with either tumor suppressor or oncogenic roles. Furthermore, this review pays particular attention to the implication of miRNAs in modulating the tumor microenvironment effectors, and how these set the basis for the discovery of novel biomarkers for early detection of oral malignancies.

## 2. Altered miRNAs in Oral Cancer

Dysregulation of miRNAs in Oral Squamous Cell Carcinoma (OSCC) and Tongue Squamous Cell Carcinoma (TSCC) has been the subject of a considerable number of scientific research studies. It has been reported that in oral cancer miR-21, miR-24, miR-31, miR-184, miR-211, miR-221, miR-222 are upregulated, while, miR-203, miR-100, miR-200, miR-133a, miR-133b, miR-138, and miR-375 are downregulated [26,27]; thus, influencing key molecular mechanisms responsible for oral carcinogenesis (Table 1).

## 3. Oncogenic miRNAs in Oral Cancer

MiRNAs with upregulated functions in oral cancer can be classified as oncogenic miRs, as they detain the ability to target and silence tumor-suppressor genes; thus, promoting cancerous-associated events such as genesis, progression, and metastasis (Figure 2).

One of the most emblematic miRNAs, miR-21, retains its capability of inducing fascination via several pathogenic effects that it can be associated with. To begin with, this miR is found to be upregulated in oral neoplasia, a dysregulation that is reported to elicit carcinogenic effects [64]. It has been demonstrated that this miRNA can reduce levels of Tropomyosin (TPM1), as well as of Phosphate and Tensin Homolog (*PTEN*) [28,29]. Functioning as a reverse effect to Phosphatidylinositol 3-Kinase (PI3K), by dephosphorylating the Phosphatidylinositol 3,4,5-trisphosphate (PIP3) to PIP2, *PTEN* is deemed as an important tumor suppressor gene. Downregulation of *PTEN* expression by miR-21 is directly responsible for hyperstimulation of the Phosphatidylinositol 3-Kinase/AKT Serine-Threonine Kinase 1 (PI3K/AKT) signaling pathway and leads to a cell’s elevated resistance to death and its uncontrollable proliferation [65].

Furthermore, bearing in mind the fact that TPM1 plays an important role in inducing cell apoptosis and migration inhibition, it is reasonably clear why a reduced level of expression may have profound consequences [66]. In another study, it is reported that miR-21 is capable of promoting perineural invasion in the oral carcinoma [67]. This finding is particularly alarming, as more researchers are associating perineural invasion with regional metastasis, and further with very poor prognosis [68]. Programmed Cell Death 4 (*PDCD4*) is a gene whose role is defined as being involved in battling against neoplastic diseases by inducing translational suppression of mRNA [69]. Taking into consideration the fact that miR-21 is found to be responsible for reducing expression of *PDCD4*, it can be affirmed that elevated levels of this oncogenic miR not only can be used as a biomarker in tumor detection, but this can also represent a definitive element in establishing the survival prognostic [28]. For instance, reduced levels of *PDCD4* promote tumor cell invasion, and this can influence the development of neck nodal metastasis in the OSCC [30]. These complications reduce survival rates of patients with oral carcinoma [70]. Moreover, studies have demonstrated that miR-21 is a determinant factor in the transition from an inoffensive leukoplakia to a malignant form [71].

MiR-24 is another strong example of the connection between neoplastic diseases and miRNAs, more specifically of the upregulation of this miR and repression of the F-Box and WD-40 domain protein 7 gene, *FBXW7*. This is critical when it comes to the uncontrolled proliferation of cancer cells, as *FBXW7* is responsible for the inactivation of several proto-oncogenes [31]. For instance, in breast cancer, it has been discovered that an *MYC* (proto-oncogene BHLH Transcription Factor; MYC) is a direct target of the FBXW7 protein, resulting in modifications of MYC protein levels and deregulation of a cell’s cycle [72]. Moreover, miR-24 can target the 3′-UTR region of *PTEN*, a mechanism that can induce resistance to chemotherapy treatment. A decrease in levels of the PTEN protein will lead to a perturbation in the PTEN/AKT pathway and activation of survival signals [73]. Furthermore, suppression of the cyclin-dependent kinase inhibitor 1B (*CDKN1B*) is accredited to miR-24. By altering the expression of the Dead-End Homolog Protein 1 (*DND1*), miR-24 assists in the proliferation of cancer cells and apoptosis escape in TSCC. Further studies are required for a definitive understanding of this mechanism [33].

In comparison to miR-24, overexpression of miR-31 is related to late tumor stages, as levels of miR-31 are found to be elevated in early stages with no metastatic nodes in OSCC [74]. The importance of miR-31 in carcinogenesis has been associated with its capability of targeting the 3′UTR region of the Factor Inhibiting Hypoxia-inducible factor (*FIH*) [35]. This factor acts as an inhibitor of the hypoxia-inducible factor (*HIF*), maintaining the balance between HIF involvement in both normoxic and hypoxic conditions. Thus, reducing the production of FIH by miR-31 will lead to activation of HIF in normoxic environments, and this will contribute to the expansion of the head and neck squamous cell carcinoma (HNSCC) [35]. Other studies have reported that miR-31 interacts with the AT-Rich Interaction Domain 1A (*ARID1A*), thereby indirectly altering gene expression levels by modifying levels of accessible chromatin. It is important to mention that miR-31 and *ARID1A* are found to be antagonistic in tumor cells, wherein the *ARID1A* gene is inhibited, thus promoting differentiation of stem cells into cancerous cells [36]. It has been widely reported that miR-31 has a vast and complex mechanism of action in cancerous cells. By targeting *Ku80*, a DNA repair gene, cancerous cell migration, and tumor genesis are increased in OSCC and ESCC mouse models [37]. As mentioned above, miR-31 is capable of modulating lipid metabolism by targeting a critical enzyme in the peroxisome, *ACOX1*, thus sustaining carcinogenesis. As miR-31 plays a specific role in β-oxidation of a large variety of fatty acids, such negative regulation of *ACOX1* may lead to accumulation of such fatty acid compounds, thereby supporting cancerous cell proliferation and migration [38]. In addition, *SIRT3* gene was validated as a new target of miR-31 in OSCC. As a consequence, miR-31-SIRT3 regulatory axis modulates not only the metabolism of OSCC cells but also improved their and aggressiveness [39].

Along with other upregulated miRNAs, miR-184 may be considered an important factor that can contribute to the pathogenesis of oral cancers. However, limited research has been conducted to demonstrate this hypothesis. It has been reported that miR-184 can have a detaining proliferative function, thus avoiding apoptosis effects [75]. In another study, a connection between long non-coding RNA urothelial cancer-associated 1 (lncRNA UCA1) and repression of miR-184 has been found; thus, this promotes proliferation and cisplatin resistance of oral squamous cell carcinoma [40].

On the other hand, miR-211 participates in enhancing carcinogenesis of OSCC by downregulating Transcription Factor 12 (*TCF12*) [41] and c-MYC expression, as well as targeting TGF β receptor II (*TGFBR2*) [42]. *TCF12* withholds suppressor activities while *TGFBR2* promotes changes in the microenvironment that favors the differentiation of neoplastic cells. In addition, reduced levels of expression of the *BIN1* gene observed in OSCC cell lines are attributed to the downregulation of the *BIN1* gene by miR-211 as it interacts with its 3′-UTR domain. As a consequence, this enhances both migration and invasion of cancerous cells. A possible explanation for such a behavior is the abnormal activation of the EGFR/MAPK pathway that regulates a variety of cellular processes such as stress responses, proliferation, differentiation, and apoptosis [43]. Furthermore, as miR-211 is reported to support dissemination and colonization of oral cancer cells, upregulation of this miRNA has been associated with poor prognosis [76].

It has been previously reported that both miR-221 and miR-222 are upregulated in about 40% of all OSCC cases, thus suggesting their primordial involvements in OSCC tumorigenesis. In particular, it has been observed that there is an inverse relationship between levels of these microRNAs and *p27/p57*, thereby suggesting that *p27/p57* are possible targets of miR-221/222 [44,45]. Furthermore, miR-222 also targets the ATP Binding Cassette sub-family G member 2 (*ABCG2*), thereby reducing its expression, and resulting in cisplatin resistance and invasive potential [46]. This enhanced migration is also promoted by targeting the BCL2 Binding Component 3 (*BBC3* or *PUMA*) gene [47].

Another relevant miR involved in oral cancer proliferation and differentiation is miR-455. Following analysis of a downstream target gene, it is reported that miR-455 has a potential ability to target the Ubiquitin Conjugating Enzyme E2 B (*UBE2B)* gene and to negatively regulate its expression [48,77]. The UBE2B enzyme plays a role in regulating DNA damage tolerance and DNA repair pathways, thereby contributing to significant increases in both tumor volume and weight in a xenograft animal model [48,77].

Overall and as described above, single miRNAs are capable of downregulating multiple tumor-suppressor genes. However, in some instances, multiple miRNAs are involved in suppressing the activity(es) of a single tumor-suppressor gene, thereby resulting in stronger inhibition. This is the case for the RAR-Related Orphan Receptor A gene (*RORA*), wherein five miRNAs are cooperatively involved in impairing the function of RORA, a master anti-tumorigenic factor in OSCC [78]. Therefore, miRNA-mRNA interactions are very complex, wherein a “gradient” of tumor-suppressor inhibition can depend on the number of targeting miRNAs.

## 4. Tumor Suppressor miRNAs in Oral Cancer

Tumor suppressor miRNAs are miRs that are downregulated in oral cancer, and these can target oncogenes, thereby influencing levels of cancerous-associated proteins (Figure 2). Under physiological conditions, miR-203 has been of particular interest due to its tumor suppressor activities. MiR-203 is simultaneously responsible for targeting and reducing the expression of the YES Proto-Oncogene 1 (Src Family Tyrosine Kinase, *YES1*) [49]. Moreover, upregulation of miR-203 in an OSCC cell line results in an antiproliferative effect. This is attributed to the ability of miR-203 to bond to the 3′-UTR region of the Phosphatidylinositol-4,5-Bisphosphate 3-Kinase Catalytic Subunit α gene (*PI3KCA*); thus, rescuing the activity of the AKT Serine/Threonine Kinase, and indirectly that of the whole signaling pathway. In addition to these antiproliferative effects, miR-203 is also capable of sensitizing tumor cells to cisplatin-induced death [50].

In a few studies focused on alterations of miR-100, it is proposed that this miRNA can be included along with other downregulated miRNAs as being involved in the pathogenesis of oral cancers. Henson and his team suggested that a reduction of miR-100 levels is related to loss of the chromosome 11q arm. This is not surprising, as genetic aberrations are common in OSCC, particularly alterations in chromosome 11. Transfection of an OSCC cell line that displays a chromosome 11q deletion with miR-100 has resulted in reduced proliferation rate and sensitivity to ionizing radiation of tumor cells, thereby supporting the involvement of miR-100the in the oral cancer development [53].

As for miR-200 family members, their involvement in the pathogenesis of oral cancer may be attributed to their dysregulation during the epithelial-mesenchymal transition, thus serving as potential markers of progression to tumor metastasis [54]. In addition, the renewal of oral cancer stem cells seems to be also regulated by miR-200 family members, with likely implications in the tumor relapse [79].

MiRNA profiling of normal tongue squamous and malignant TSCC and malignant tissues highlights upregulation of both miR-133a and miR-133b in normal tongue squamous cells compared to those of malignant cells. Thus, it is proposed that miR-133a and miR-133b function as proliferative and apoptosis regulators [79]. This observation is further supported by the transfection of TSCC cell lines with miR-133a and miR-133b. Furthermore, a reduction of proliferation rate is observed, and the association between these miRNAs and Pyruvate Kinase M1/2 (*PKM2*) is demonstrated [55]. Specifically, increased *PKM2* expression is observed in TSCC with reduced miR-133 levels. As PKM2 is expressed in a dimeric form in cancer cells, this could sustain tumor cell development, even under conditions of low glucose and oxygen environment. This could partially explain the tumor-suppressor roles of miR-133a and miR-133b [55]. Additionally, miR-133a seems to control levels of Glutathione s-Transferase P1 (*GsTP1*), but additional studies should be conducted to confirm this hypothesis. The hypothesis is based on the observation that low expression of miR-133a leads to elevated levels of GsTP1 that would harm a cell’s anti-proliferative mechanisms [56].

Among a group of 50 genes, the gene encoding for the G Protein Subunit α I2, *GNAI2*, serves as a target for miR-138. This finding has improved our understanding of the mechanisms involved in the carcinogenesis of TSCC. Modifications of the GNAI2 protein have been detected in other malignancies as well, such as ovarian cancer, wherein an abundance of GNAI2 leads to an uncontrolled proliferation of cells [80]. Similar findings have been observed in oral cancer, whereby downregulation of miR-138 may lead to increased levels of *GNAI2* along with substantial implications on a cell’s cycle and apoptosis [57]. Furthermore, this damage is accentuated by the capacity of miR-138 to induce metastasis, and for this reason, it may serve as a potential prognostic marker of aggressive disease [81]. Furthermore, it has been reported that miR-138 influences levels of the Ras homolog family member C oncogene (*Rhoc*), a factor that contributes to cancer stem cell formation in the HNSCC [58].

Several studies have been conducted to assess the role(s) of miR-375 on the onset, development [59], and invasion [62] of oral cancers. It has been assumed that miR-375 is downregulated in oral carcinoma, and experimental results have supported this premise. Furthermore, using aggressive metastatic cell lines, dissemination of cancerous cells among other healthy tissues has been demonstrated. This noxious characteristic of miR-375 is associated with dysregulations of the platelet-derived growth factor-A (*PDGF-A*) [60]. Moreover, the Solute Carrier Family 7 Member 11 gene, *SLC7A11*, seems to be a target of miR-375as it modulates levels of this Cysteine-Glutamate transporter, with implications for the unpredictable behaviors of malignant cells [62]. Another target of miR-375 seems to be the cellular inhibitor of PP2A (*CIP2A*), as it has considerable numbers of interactions with MYC. In tumorous tissues, it has been observed that there is an inverse correlation between higher levels of miR-375 and lower levels of CIP2A, thereby highlighting the importance of miR-375 in the pathogenicity of the oral cancers [63].

## 5. Modulation of Oral Cancer Tumor Microenvironment Components by miRNAs

In the past few years, it has been demonstrated that the tumor microenvironment (TME) is strongly interconnected with tumor cells, stimulating both initiation and progression the of cancer [82]. TME of oral cancers is composed of two major components, the extracellular matrix (ECM) and the cellular components such as fibroblasts, neurons, endocrine cells, adipocytes, and cells of the immune system [83]. It has been discovered that in pathological conditions, TME becomes more than an accomplice in oral cancer’s development and spread [84]. Therefore, it is of utmost importance to understand the processes that occur at this level [84]. It is well known that the first response to any dysregulation is that of the immune system and its responsibility in maintaining tissue homeostasis [85]. In tumorigenesis, major changes occur in macrophage behavior, thereby resulting in the large-scale conversion of antigen-presenting macrophages (M1) to type II cytokine-producing macrophages (M2). It seems that environmental changes such as hypoxia may be responsible for the transition between anti-carcinogenic M1 to pro-carcinogenic M2 [86]. It is reported that miR-31, one of the most active oncogenic miR in HNSCC, detains the ability to target *FIH,* and hence induces hypoxia [35]. Expression levels of miR-21 in stromal cells are found to be higher than those in cancerous squamous cells [87]. Thus, the miR-21 may serve as a marker for tumor-associated myofibroblast identification [87]. Furthermore, among other upregulated miRNAs, miR-21 seems to stimulate the Transforming Growth Factor β 1 (TGFB1) pathway, as well as protein metabolic processes that support the stromal myofibroblast differentiation [88]. It has been observed that higher numbers of myofibroblasts facilitate malignant cell vascular-lymphatic invasion and lymph node metastasis, thereby decreasing survival rates of patients with OSCC [89]. More interestingly, miRNAs can also modulate both migration and invasion of oral cancerous cells via targeting specific genes of microenvironment components. A suitable example is that of miR-148a. This miR is reported to target the key gene Wnt Family Member 10B, *WNT10B*, in Cancer-Associated Fibroblasts (CAFs), resulting in an uncontrollable spread of malignant cells [90]. In TSCC and other cancers, CAFs provide a proper environment for proliferation and differentiation of a tumor due to their capacities to secrete not only cytokines and chemokines but also growth factors [91,92]. In another study, the critical role of miR-124 in oral carcinoma development has been investigated as it targets Chemokine (C-C motif) Ligand 2 (*CCL2*) and Interleukin 8 (*IL8*) in both CAFs and oral cancer cells [93].

Lymphangiogenesis is the formation of lymphatic vessels from pre-existing lymphatic vessels, and recent studies have suggested that miRNAs are also involved in this process. The most suitable example is that of miR-126. It has been reported that downregulation of miR-126 leads to increased gene expression of the Vascular Endothelial Growth Factor A, *VEGF-A*, and further of oral cancerous angiogenesis and lymphangiogenesis [94]. A similar example is that of miR-300, a miRNA that is regulated by a cysteine-rich protein, WNT1-inducible Signaling Pathway Protein 1 (WISP-1/CCN4), a matrix-related protein that belongs to the CCN family [95]. It is reported that WISP-1 is responsible for the downregulation of miR-300, thereby contributing to enhanced production of the Vascular Endothelial Growth Factor C (VEGF-C). Consequently, the formation of OSCC lymphatic vessels will be boosted and will favor the development of cancer’s metastasis [95]. On the other hand, WISP-1 also regulates VEGF-A [96]. Therefore, it would be interesting to determine if there is a connection between WISP-1 and miR-126 [96].

Tumorigenesis of oral cancer is stimulated by the capability of miR-320 to target the 3′-UTR of Neuropilin 1 (*NRP1*) [97]. NRP1 functions as a promoter of proliferation, survival, and migration/invasion of tumor cells and endothelial cells in various tumor type [98,99,100]. Therefore, miRNAs, such as miR-320, can serve as innovative targets for anti-cancerous drugs [97]. Recently, a novel drug, niclosamide, has been identified as it is reported to reduce vasculogenic mimicry (VM) by influencing expression levels of miR-124 [101]. VM is best described as the ability of aggressive cancers to spontaneously generate blood vessels without the presence of endothelial cells. This will ensure the availability of sufficient blood supply for oral cancer development and metastasis [102]. The drug niclosamide is capable of inducing anti-tumor effects by up-regulating miR-124 and thereby inhibiting VM [101].

The above examples of evidence highlight the importance of miRNAs in TME regulation. Moreover, the fact that miRNAs influence the conversion rate and migration of tumor stem cells has not been overlooked by scientists. In OSCC, reduced expression levels of miR-204 are associated with a poorer survival rate because its downregulation seems to be involved both in epithelial-mesenchymal transition and cancer cells stemness. These findings can be explained by the ability of miR-204 to target the Snail Family Transcriptional Repressor 2 (*SLUG/SNAI2*) and the SRY-Box Transcription Factor 4 (*SOX4*) genes and limit the uncontrollable behavior of cancer stem cells [103].

Several studies have attempted to counteract the aggressive behavior of oral cancer stem cells by targeting specific miRs. Such examples are the deliberate increases of miR-218 and miR-145 expression to downregulate the BMI1 Proto-Oncogene Polycomb Ring Finger (*BMI1*) and CD44, respectively [104,105]. A listing of the major miRNAs involved in TME development in oral cancer pathogenesis is presented in Table 2.

## 6. miRNAs as Potential Biomarkers for Oral Cancer

Differential expression of miRNAs between malignant cells and normal cells, as well as their detection in liquid biopsies such as serum, plasma, or saliva, in free or vesicular form, render miRNAs as desirable biomarkers for oral cancer diagnosis and prognosis [109,110] Based on recent studies, there are four sources of circulating miRNA (Figure 3): miRNAs resulting from the ‘enclosing with membrane’ process, which culminates in exosomes active secretion of the microRNA [111,112]; miRNAs lacking the protection of membrane vesicles, but are in association with either RNA-binding proteins (AGO 1–4) or high-density lipoproteins (HDL) [113,114]; miRNAs resulting via the export of apoptotic bodies [115]; and those miRNAs derived through exocytosis as shedding vesicles (SV) [116,117,118].

Even though it has earlier been hypothesized that these extracellular genetic products, circulating miRNAs, have no real function, studies have revealed that expulsion of microRNAs into the extracellular environment may be a regulated process with implications for the cell-cell communication [119]. In addition, miRNAs, particularly exosomal miRNAs, are found to be involved in the hallmarks of cancer, regulating proliferation, apoptosis, metastasis formation, and invasion of breast cancer cells [120]. For example, miR-21-rich tumor-derived exosomes exert considerable influence on the invasiveness and migration of OSCC cells. Such exosomes are highly abundant in hypoxic regions of solid tumors, and they may even interact with normoxic cells and increase their aggressiveness and metastatic potential [121]. It has been reported that packaging of specific miRNAs in different extracellular vesicles in head and neck carcinoma is not only highly correlated with cell type, but they can also serve as critical vectors for either promoting or antithetically suppressing the development of malignant lesions [122].

It is of particular interest that miRNAs are found to be resistant against RNaseA digestion and the grim condition of serum. This explains at least partially the formidable stability of this genetic material in the extracellular environment [123]. Another advantage of using the serum as a sampling medium is the reproducibility and consistency of this medium. Several studies have revealed that different miRNAs such as miR-483, miR-9, or miR-21 can be identified in the serum and that their levels can be easily correlated with the prognostic of OSCC [124,125,126]. Moreover, similar outcomes are reported in saliva, as his environment has appreciable quantities of miRNAs. The fact that a small amount of this saliva medium is required to determine levels of miRNAs, circulating miRNA extracted from saliva can serve as ideal for noninvasive oral cancer detection. It has been reported that miR-31 is a viable diagnostic biomarker candidate for the development of oral cancer. When comparing expression levels of miR-31, both in plasma and saliva, in patients with either OSCC or oral verrucous leucoplakia (OVL) and healthy controls, it is found that miR-31 levels are higher in OSCC patients compared to those in OVL patients and controls [127]. Furthermore, when determining miR-31 levels in saliva of OSCC patients before and after primary tumor resection, a significant reduction in miR-31 is detected after surgery, suggesting that it is the primary tumor is the source of circulating miR-31 [127]. As both plasma and saliva have comparable levels of miR-31 expression in OSCC patients with incipient and advanced disease, this points to the fact that the oral fluid is well-suited for the diagnosis of insidious malignant lesions [127]. Similarly, other studies have proposed the use of salivary miRNAs as markers for the progression of premalignant lesions, such as low-grade dysplasia, to OSCC. In addition to offering information on oncogenic miRNAs expression, salivary tumor suppressor miRNAs also serve as valuable markers for protection for malignant transformation [128]. It has been reported that high yields of miRNA can be obtained from as little as 200 μL of input saliva using QIAzol reagent, followed by common solid-phase microtube chromatography for the enrichment of miRNAs [129]. Interestingly, both fresh and archived frozen saliva offer comparable amounts of high-quality miRNAs [129]. More recently, significant progress has been made in assessing expression levels of miRNAs in saliva using nucleic acid detection with CRISPR/Cas12a along with a simple design of paper-based strip testing [130]. This technical advance serves as a very attractive alternative to the classical quantitative analysis by RT-qPCR, which is more labor-intensive and costly. This novel method, referred to as point-of-care testing (POCT), has been used for ultrasensitive detection of miR-31 extracted from 150 µL of saliva collected from OSCC patients [130].

Few clinical studies have been conducted to confirm all advantages of using miRNAs as novel biomarkers. An example is miR-27b, found to be upregulated both in plasma and oral tumor tissue, which has been deemed as a better biomarker than that of proteins [131]. On the other hand, microarray and RT-PCR analyses of oral carcinoma and normal control tissue samples have highlighted differences between miRNAs expression levels [132]. It is reported that miR-196b, miR-1237, and miR-21 are found to be up-regulated, while miR-190, miR-204, and miR-144 are found to be down-regulated [132]. Furthermore, miR-196b, deemed as a highly capable marker for predicting survival rates of patients, is found to be highly up-regulated in a low-survival group, and it gradually decreases with increasing survival rates [132]. A similar finding is reported for miR-376a and miR-204 [132]. Furthermore, a combinatory potential of miRNA expression ratios in disease prediction has been evaluated, and it is observed that patients expressing a miR-196a/miR-204 ratio < 1 have increased morbidity [132].

Efforts to support the use of miRNAs as diagnostic biomarkers have been ongoing. In a recent comparative analysis of data in The Cancer Genome Atlas (TCGA), miRNA expression of tumor samples and normal samples along with corresponding clinical information retrieved from HNSCC patients, it is reported that no single miRNA is found potent enough to differentiate cancer samples from healthy samples [133]. However, if a combination of three or more miRNAs is used, the prediction accuracy would drastically improve. For instance, expression levels of mir-383, mir-615, and mir-877 in serum of HNSCC patients versus healthy controls have allowed for the identification of patients with 89.3% sensitivity and 98.9% specificity [133]. Thus, miRNAs may serve as potential diagnostic biomarkers for cancer.

The outset of oral cancer, a multifactorial long-term disorder, can be simply described as the moment when genetic alterations occur in cytogenesis and transcription processes of keratinocyte stem cell genes [134]. It is important to identify the presence of genetically modified cells during the early stages of oral cancer, as this is not often detected at clinical levels. However, aberrant keratinocytes can participate in the formation of some oral disorders, with a notable susceptibility of evolution to the oral cancer [135]. Leukoplakia, a white patch in oral mucosa, not attributed to any other dysfunction, is a suitable example of such oral disorder [136]. Studies have revealed that there is a strong likelihood, even up to 37.5%, of leukoplakia to evolve into a malignant stage, denoting that it can be deemed as an important and monitorable risk factor [137]. Therefore, it is highly important to identify a potent indicator to distinguish between progressive and non-progressive leukoplakia. At this time, there are three miRNAs, including miR-21, miR-181b, and miR-345, that serve as likely candidates as they are found to be over-expressed both in OSCC and progressive dysplasia [138]. Remarkably, levels of expression of these three miRNAs can be correlated with the evolution towards cancerous lesions; thus, demonstrating a higher association between miRNAs and the carcinogenesis [138]. In another study, it is reported that there are about 14 essential genes involved in oral cancer genesis from leukoplakia, including Signal Transducer and Activator of Transcription 5B (*STAT5B*) and Epidermal Growth Factor Receptor (*EGFR*), and eight miRs, such as miR-549, miR-205, and miR-21 [71]. Although further studies are required, it seems that miRNAs are involved in the transition to a malignant stage due to their capabilities of inducing genomic instability. From a practical point of view, it is worthy to consider the use of miRs as markers for early detection of cancerous lesions, as a reliable diagnostic tool for reducing mortality caused by oral cancer.

## 7. Concluding Remarks and Future Perspective

The discovery of miRNAs serves as a turning point in the ongoing process of understanding the malignant phenotype of neoplastic cells. Several altered miRNAs are likely to be involved in the initiation and the development of carcinogenesis, either through increased expression levels of oncogenes or by constraining the expression of tumor suppressor genes. However, more importantly, miRNAs offer opportunities for increased attention towards precocious early detection and less on the late treatment of oral cancers. Based on current technologies, early screening is a cheap and effective approach to reduce cancer mortality rates in the human population, and miRNAs can serve as key players in achieving better clinical outcomes in oral cancer patients. For the long term, the discovery of new non-invasive methods for detecting oral cancer, a simple visit to a dentist would aid tremendously in the evaluation and diagnosis of many oral pathological conditions. However, it would be interesting to know which are the most incipient stages of a malignant process and how early these can be detected to prevent the onset of oral tumor development and progression. More importantly, if such means of early detection of biomarkers, such as miRNAs, reflect the burst of a malignant neoplastic event, can these offer information on the localization of a particular site when a premalignant lesion is not visible during the regular clinical evaluation? Such findings would offer a clinician with valuable information on whether the patient is going to proceed with a therapeutic intervention for developing oral cancer. In addition to the potentials of miRNAs as biomarkers, these miRNAs can also be regarded as targets for therapy, either by decreasing elevated levels of oncomiRs or by increasing expression levels of tumor suppressor miRNAs. To this end, major efforts are needed to establish whether one or more miRNAs must be targeted for sustained tumor regression and if such a strategy can be used as a monotherapy or as an adjuvant therapeutic approach. Furthermore, the time frame of using miRNA therapy is equally important to be determined, as either a short or prolonged treatment can result in tumor relapse or significant side effects.

## Figures and Tables

**Figure 1 genes-12-01989-f001:**
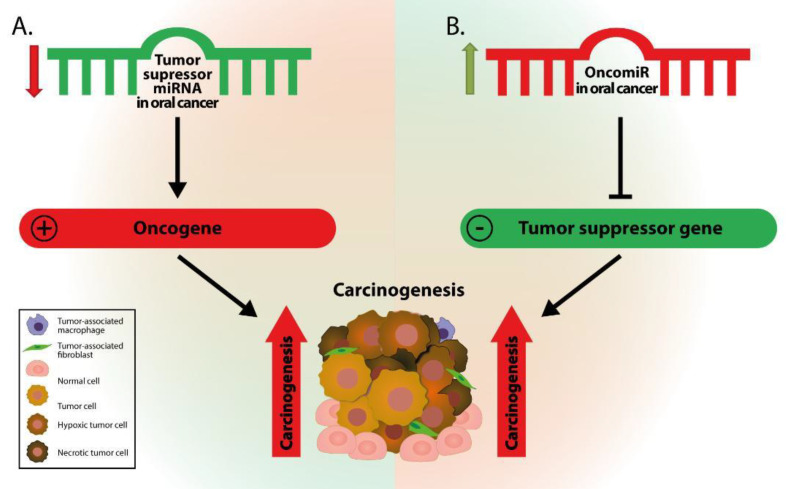
The impact of miRNA dysregulation on gene expression and, further, on carcinogenesis. (**A**) Downregulation of tumor suppressor miRNAs results in increased expression of oncogenes, and further to elevated levels of tumor-promoting proteins, leading to increased carcinogenesis. (**B**) Upregulation of an oncogenic miRNA (OncomiR) leads to repression of tumor suppressor gene expression and reduced expression of tumor suppressor proteins that further promote carcinogenesis.

**Figure 2 genes-12-01989-f002:**
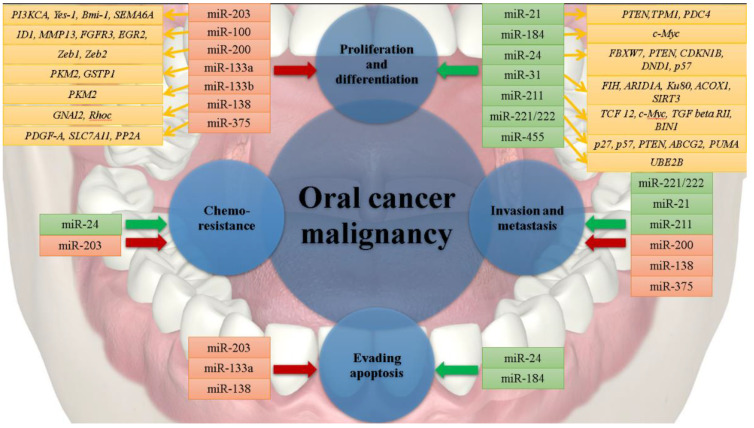
Associations between miRNAs and their target genes in oral cancer. As illustrated, upregulated miRNAs (in green-colored boxes) and downregulated miRNAs (in salmon-colored boxes) can have significant effects in controlling various levels of cell homeostasis, by either reducing expression levels of tumor suppressor genes or by increasing expression levels of oncogenes or tumor suppressor genes (in yellow-colored boxes). As a result, normal cells evolve progressively to a neoplastic state, achieving special properties known as “the hallmarks of cancer” (dark-green and dark-red arrows).

**Figure 3 genes-12-01989-f003:**
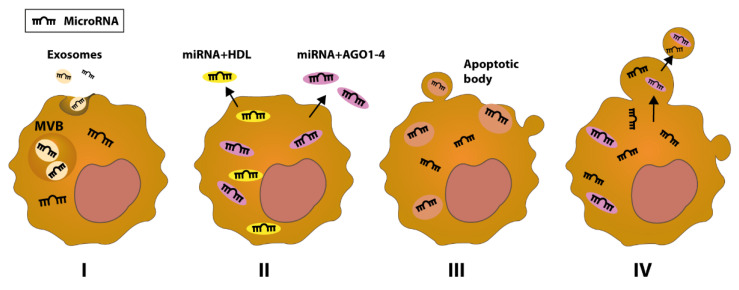
A simplified diagram of the formation processes and sources of circulating miRNAs. (**I**) MiRNAs are released from cancer cells within exosomes by the active mechanism of exocytosis. Multivesicular bodies (MVB) merge with the cellular membrane, releasing exosomes. (**II**) Vesicle-free miRNAs represent most of the circulating microRNAs. The association between miRNAs and RNA-binding proteins such as AGO 1–4 maintains the stability of associated miRNAs against nuclease and protease. The complex can be either released into the extracellular space or can be loaded into shedding vesicles. MiRNA can also be attached to high-density lipoproteins (HDL). (**III**) MiRNA can be released via apoptotic bodies containing diversified cellular organelles including miRNAs. (**IV**) MiRNA can be delivered into circulation, as they are enclosed with a cellular membrane forming a shedding vesicle (SV).

**Table 1 genes-12-01989-t001:** The most important dysregulated miRNAs and their impact on oral cancer pathogenesis.

microRNA	Level of Expression	Biological Influence on Cancerous Cells	Target Genes	Area of Assessment	Biomarker Role	References
miR-21	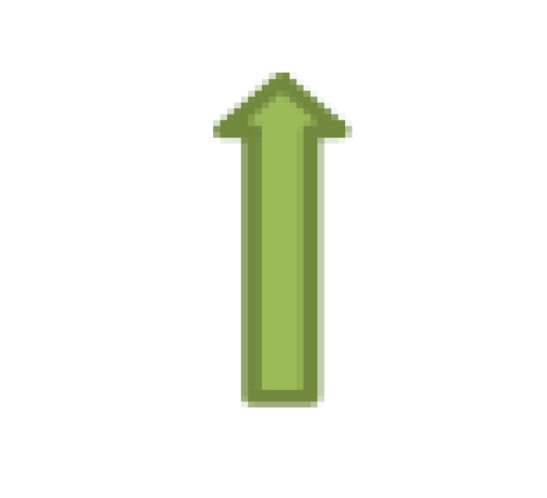	Proliferation Invasion and Metastasis	*PTEN, TPM1, PDC4*	Tumor tissue Saliva	Diagnostic and prognostic	[28,29,30]
miR-24	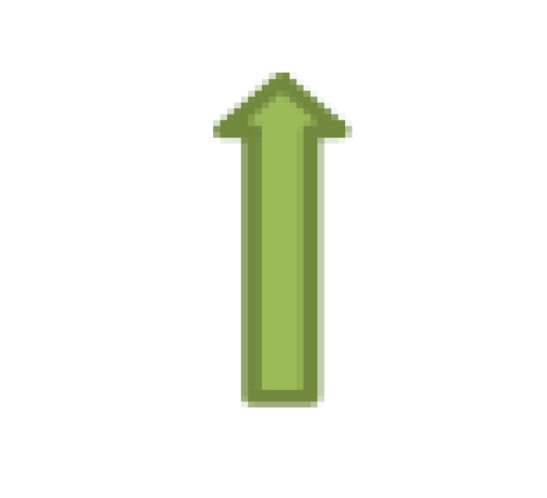	Proliferation Resistance to chemotherapy Antiapoptotic	*FBXW7, PTEN, CDKN1B, DND1, CDKN1C*	Tumor tissue Plasma Saliva	Diagnostic	[31,32,33,34]
miR-31	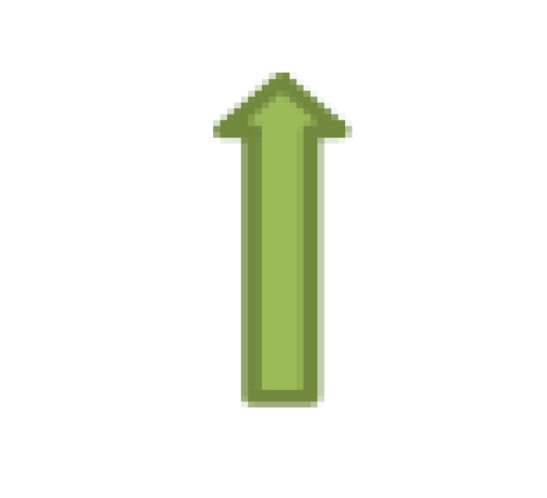	Proliferation and Differentiation Invasion and Metastasis	*FIH, ARID1A, Ku80, ACOX1, SIRT3*	Tumor tissue Saliva	Diagnostic and prognostic	[35,36,37,38,39]
miR-184	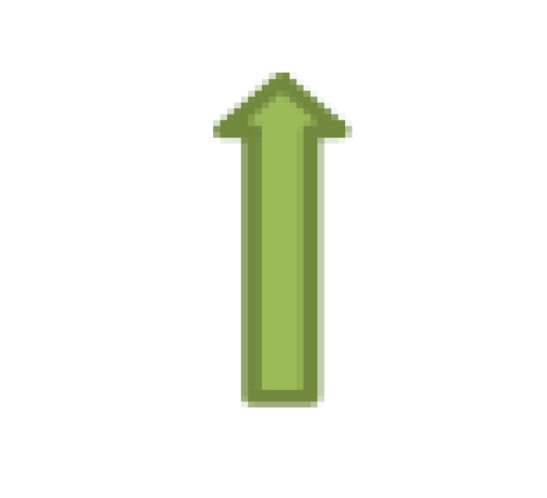	Proliferation Antiapoptotic	*c-MYC*	Tumor tissue Saliva	Diagnostic	[40]
miR-211	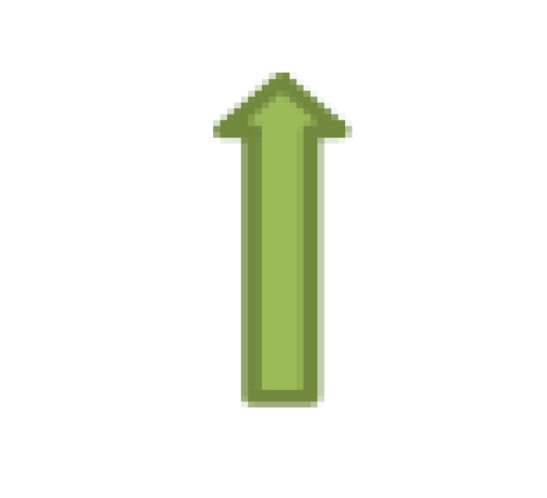	Proliferation and Differentiation Invasion and Metastasis	*TCF12, c-MYC, TGFBR2, BIN1*	Tumor tissue	Diagnostic and prognostic	[41,42,43]
miR-221, miR-222	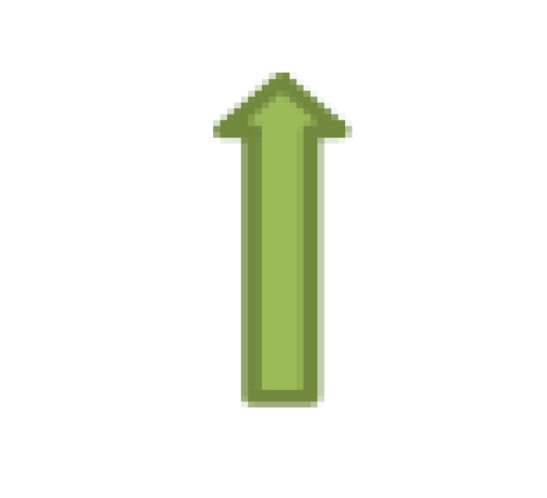	Proliferation Invasion and Metastasis	*CDKN1B, CDKN1C, PTEN, ABCG2, PUMA*	Tumor tissue Saliva (only miR-221)	Diagnostic	[44,45,46,47]
miR-455	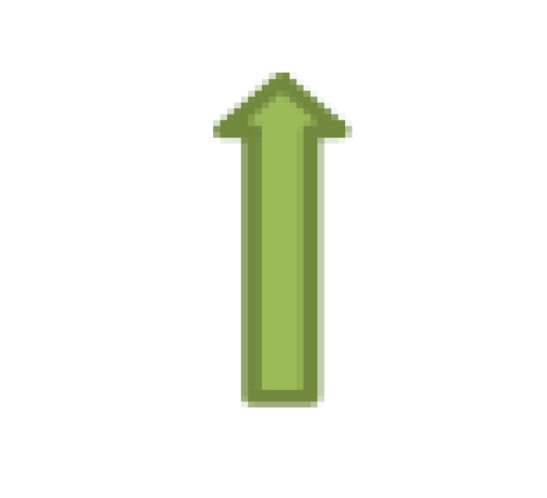	Proliferation and Differentiation	*UBE2B*	Tumor tissue	Diagnostic and prognostic	[48]
miR-203	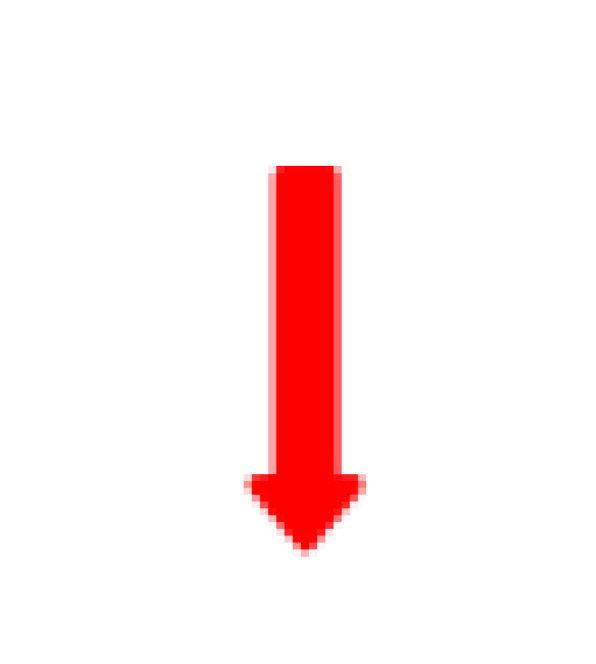	Proliferation Antiapoptotic Resistance to chemotherapy	*PI3KCA, YES-1, BMI-1, SEMA6A*	Tumor tissue Saliva (so far only in non-cancerous cells)	Diagnostic and prognostic	[49,50,51,52]
miR-100	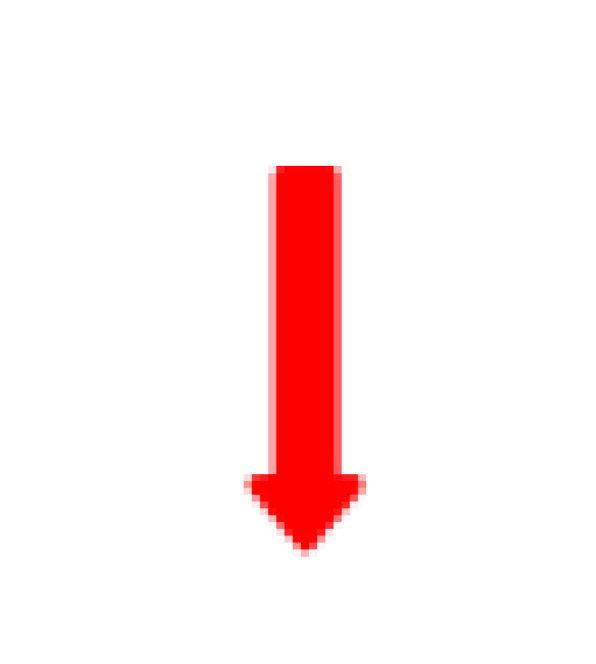	Proliferation and Differentiation	*ID1, MMP13, EGR2, FGFR3*	Tumor tissue	Diagnostic	[53]
miR-200	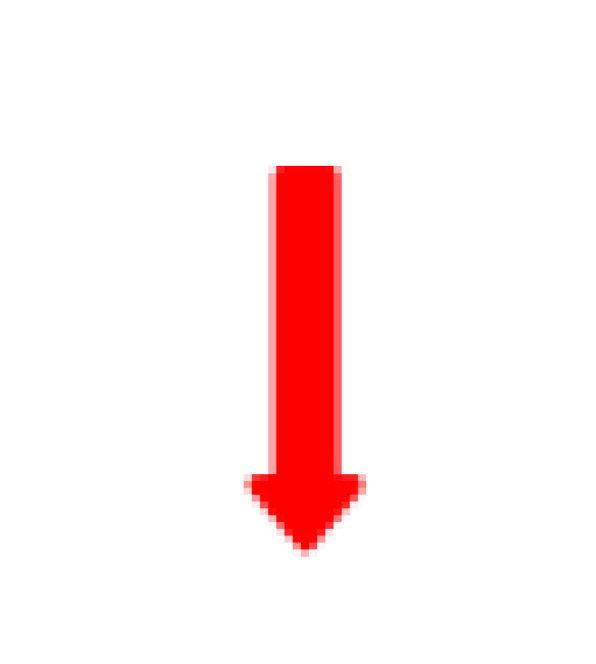	Differentiation Invasion and Metastasis	*ZEB1, ZEB2*	Tumor tissue Saliva oral rinse	Diagnostic and prognostic	[54]
miR-133a	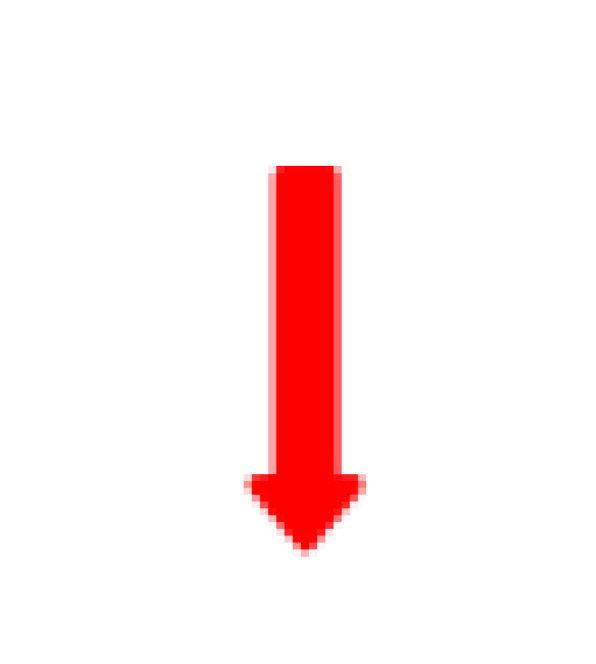	Proliferation and Differentiation Antiapoptotic	*PKM2, GSTP1*	Tumor tissue	Diagnostic	[55]
miR-133b	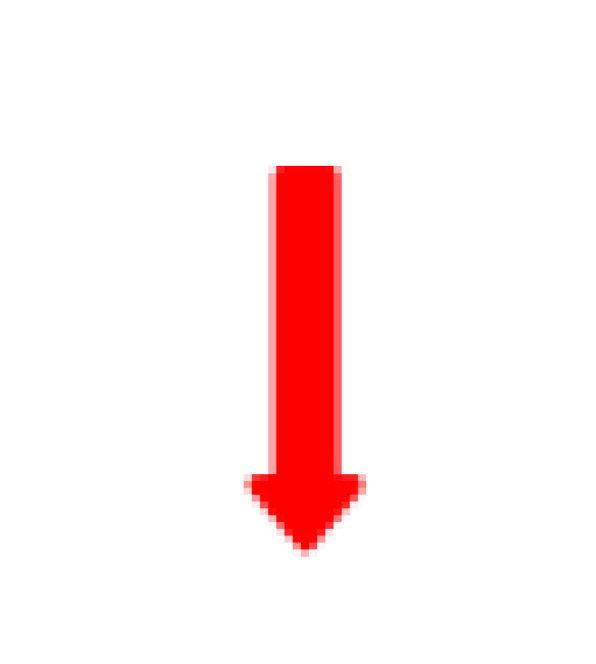	Proliferation	*PKM2*	Tumor tissue	Diagnostic	[55,56]
miR-138	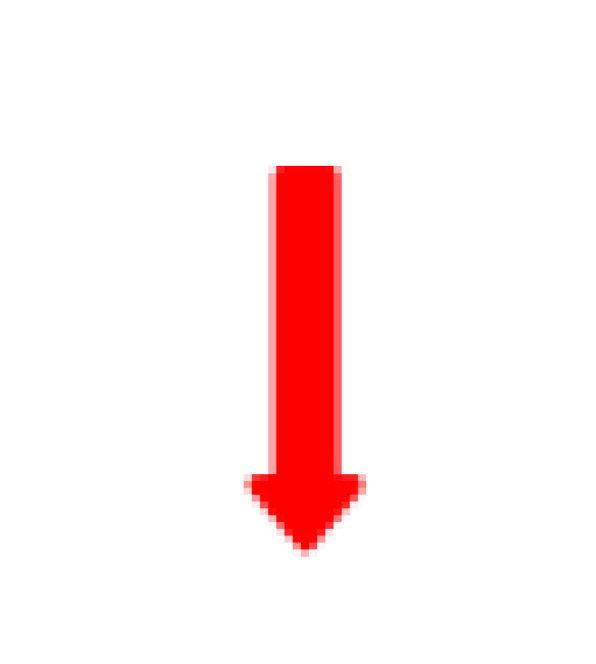	Proliferation Antiapoptotic Invasion and Metastasis	*GNAI2, Rhoc*	Tumor tissue	Diagnostic	[57,58]
miR-375	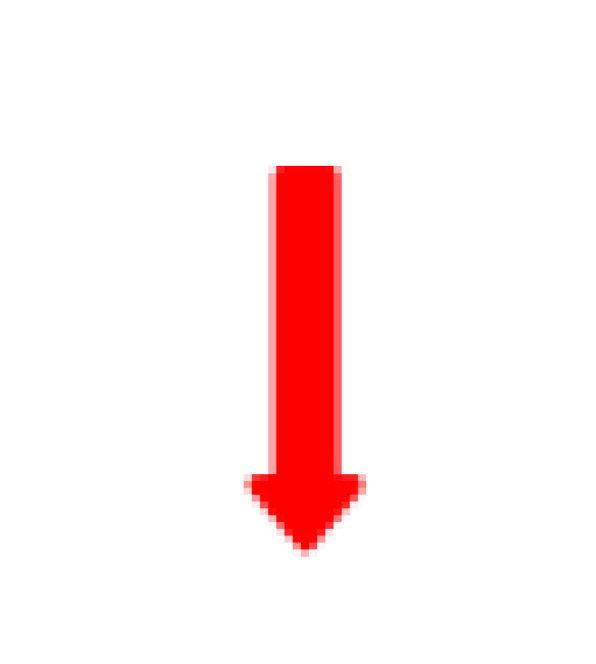	Proliferation Invasion and Metastasis	*PDGF-A, SLC7A11, CIP2A*	Tumor tissue Saliva	Diagnostic	[59,60,61,62,63]

**Table 2 genes-12-01989-t002:** Relevant examples related to the altered microRNA affecting tumor microenvironment (TME) components.

microRNA	Target Gene	Impact on Oral TME	References
miR-21	*TGFB1*	Myofibroblast differentiation	[88]
miR-148-a	*WNT10B*	Cancer-associated fibroblats (CAFs) proliferation	[90]
miR-124	*CCL2, IL8*	CAFs proliferation and migration	[93]
miR-126	*VEGF-A*	Angiogenesis and Lymphangiogenesis	[94]
miR 300	*VEGF-C*	Lymphangiogenesis	[95]
miR-320	*NRP1*	Angiogenesis	[97]
miR-124	*STAT3*	Increase of Vasculogenic mimicry	[101]
miR-204	*SLUG, SOX4*	Epithelial-mesenchymal transition (EMT), Stemness features	[103]
miR-218	*BMI1*	Stemness features	[104]
miR-145	*CD44*	Stemness features	[105]
miR-200	*ZEB1, ZEB2*	EMT	[54]
miR-153	*SNAI1, ZEB2*	EMT	[106]
miR-639	*FOXC1*	EMT	[107]
miR-143, miR-145	*Activin A*	EMT	[108]

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
