# Peer review of "The Connection between MicroRNAs and Oral Cancer Pathogenesis: Emerging Biomarkers in Oral Cancer Management"

_genes, 2021, doi:10.3390/genes12121989_

Round 1

Reviewer 1 Report

This is an interesting review illustrating the aberrances of miRNAs and their targets in OSCC. Authors also conceptualize the detection of aberrant miRNAs in body fluid, especially in saliva, for the development of biomarkers. The cited references should be more updated and the following concerns should be addressed.

  1. miR-21 is a well-known hypoxia-induced miRNA in OSCC (Cancer Res, 2016), which is also abundant in exosomes that may mediate metastasis by affecting neighboring cells. Such concept needs to be emphasized.
  2. In fact, many important targets of miR-31 are not reviewed in this article. They are Ku80 (Int J Cancer, 2015), ACOX1 (Theranostics, 2018) and Sirt3 (Cancer Lett, 2019). By means of targeting, miR-31 play pluripotent roles in OSCC by modulating DNA repair and metabolisms in addition to hypoxia and stemness.
  3. One more important target of miR-211, the BIN1 which modulates EGFR cascade (J Cell Biochem, 2019), is not reviewed in this paper.
  4. Arguments on the internal control of saliva persist. Thus, to validate salivary diagnosis using qPCR/miRNA analysis as liquid biopsy still has long way to go. However, a recent study demonstrating the power of paper-based strip via Crispr/Cas12a approach for rapid salivary detection could be a promising resolution to this issue (Analytical Chem, 2020).
  5. In line 165, miR-221 should be miR-211.

Reviewer 2 Report

RThis manuscript “The connection between microRNAs and oral cancer pathogenesis: emerging biomarkers in oral cancer management” focuses on the function of the relevant microRNAs in oral cancer development and progression, as well as their effects on microenvironment to support the potential of microRNAs as non-invasive biomarkers for oral cancers. They summarized recent literatures and categorized the relevant microRNAs based on their function, such as “oncogenic”, “tumor suppresser” and “modulators of tumor microenvironment” for oral cancers. In my opinion, this is a comprehensive review on a hot topic: non-invasive biomarkers for early detection of oral cancer. In particular, they mapped relevant microRNAs based on their detailed functions, including proliferation and differentiation, chemo-resistance, invasion and metastasis, evading apoptosis. A few suggestions are listed below:

1) Based on Figure 2, the authors categorized the relevant microRNAs based on their function. It will be better to follow this category to organize the introduction/summary of the microRNAs in each section (3,4,5).

2) According to the title, we will expect to get more insights about available evidences supporting microRNAs as potential effective biomarkers in oral cancer management (section 6).

3) Few relevant studies are missing. Here are some examples:

Lai et al., MiR-31-5p-ACOX1 Axis enhances tumorigenic fitness in oral squamous cell carcinoma via the promigratory prostaglandin E2. Theranostics. 2018;8:486–504.

Zheng et al., MicroRNA-transcription factor network analysis reveals miRNAs cooperatively suppress RORA in oral squamous cell carcinoma. Oncogenesis. 2018;7:79. doi: 10.1038/s41389-018-0089-8.

Cheng et al., Up-regulation of miR-455-5p by the TGF-β-SMAD signalling axis promotes the proliferation of oral squamous cancer cells by targeting UBE2B. J Pathol. 2016 Sep; 240(1):38-49.

4) Moderate English changes may be required. Few sentences/wording are listed here:

  Line 17-19; Line 33-34; Line 40; Line 95-96; Line 291; Line 413-415.

Please go through the whole manuscript and correct some improper phrase.

Round 2

Reviewer 1 Report

This paper has been sufficiently improved.